# Functional Gene Clusters in Global Pathogenesis of Clear Cell Carcinoma of the Ovary Discovered by Integrated Analysis of Transcriptomes

**DOI:** 10.3390/ijerph17113951

**Published:** 2020-06-02

**Authors:** Yueh-Han Hsu, Peng-Hui Wang, Chia-Ming Chang

**Affiliations:** 1Department of Obstetrics and Gynecology, Taipei Veterans General Hospital, Taipei 112, Taiwan; yhhsu7@vghtpe.gov.tw (Y.-H.H.); phwang@vghtpe.gov.tw (P.-H.W.); 2School of Medicine, National Yang-Ming University, Taipei 112, Taiwan; 3Institute of Clinical Medicine, National Yang-Ming University, Taipei 112, Taiwan; 4Department of Medical Research, China Medical University Hospital, Taichung 440, Taiwan; 5Female Cancer Foundation, Taipei 104, Taiwan

**Keywords:** ovarian clear cell carcinoma, microarray, ribosomal protein, eukaryotic translation initiation factors, lactate, prostaglandin, proteasome, insulin-like growth factor

## Abstract

Clear cell carcinoma of the ovary (ovarian clear cell carcinoma (OCCC)) is one epithelial ovarian carcinoma that is known to have a poor prognosis and a tendency for being refractory to treatment due to unclear pathogenesis. Published investigations of OCCC have mainly focused only on individual genes and lack of systematic integrated research to analyze the pathogenesis of OCCC in a genome-wide perspective. Thus, we conducted an integrated analysis using transcriptome datasets from a public domain database to determine genes that may be implicated in the pathogenesis involved in OCCC carcinogenesis. We used the data obtained from the National Center for Biotechnology Information (NCBI) Gene Expression Omnibus (GEO) DataSets. We found six interactive functional gene clusters in the pathogenesis network of OCCC, including ribosomal protein, eukaryotic translation initiation factors, lactate, prostaglandin, proteasome, and insulin-like growth factor. This finding from our integrated analysis affords us a global understanding of the interactive network of OCCC pathogenesis.

## 1. Introduction

Approximately 85–90% cases of ovarian cancers are those of epithelial ovarian cancer (EOC), and the majority of them occurred in the postmenopausal women with a peak incidence at age of 56–60 years [1,2,3]. Traditionally, ovarian cancer has been considered a “silent killer” that is asymptomatic until it reaches a far advanced stage, which presents various obvious symptoms or signs secondary to massive ascites and wide-spread intraperitoneal carcinomatosis [4]. There are several histologic types of EOC, including serous, endometrioid, clear cell, mucinous transitional, and undifferentiated carcinomas. In clear cell type about 50% of patients were classified as the International Federation of Gynecology and Obstetrics (FIGO) stage I disease [5]. However, it is the most chemo-resistant and is associated with a poor prognosis among those subtypes [6,7]. 

Besides an early stage being the defining characteristic of clear cell carcinoma of the ovary (ovarian clear cell carcinoma (OCCC)), OCCC often occurs in younger women and shows worse prognosis compared to the similar stage of other histotypes of EOC, such as high-grade serous carcinoma (HGSC) [8,9,10]. This is primarily attributable to the relative resistance of OCCC to platinum drugs. Platinum-based chemotherapy presents an efficacy of 70–80% for HGSC but 20–50% for OCCC [8]. 

Although the mechanisms underlying the chemoresistance and carcinogenicity of OCCC remain unclear, the genetic changes that may be involved in the pathways of OCCC have been extensively studied. For example, Zhou et al. discovered that OCCC has 199 significantly different expression genes compared with HGSOC *(p* < 0.05); they also discovered those genes to be associated with functions such as apoptosis, cell cycle, repair of DNA damage, and the PI3K pathway. In particular, they noted that within the PI3K pathway, there were 164 differentially expressed genes (DEGs) [9]. Another study demonstrated the different pathways in the suppression of the proliferation of OCCC- and HGSC-derived cells, with the former being mediated by inhibition of the calcium-dependent protein copine 8 (CPNE8) and the latter being mediated by inhibition of the transcription factor basic helix-loop-helix family member e 41 (BHLHE41) [11]. Additionally, the cause of molecular changes in OCCC is often associated with AT-rich active domain 1A (ARID1A) mutations [12,13,14]. Another frequent gene change in OCCC is an activation of mutations of the phosphatidylinositol-4,5-diphosphate 3-kinase catalytic subunit alpha (PIK3CA) gene, suggesting that the PI3K-AKT-mTOR pathway can be a potential targeting site [15,16].

However, there was no integrated analysis of OCCC with transcriptomes currently. Currently, research of OCCC is based on DNA microarrays as the main research method used to identify DEGs, but usually the case numbers of these studies were limited, which has resulted in few statistically significant genes being found with statistical significance and did not identify the overall pathophysiology of OCCC. Therefore, we conducted an integrated analysis with the transcriptome datasets downloaded from the public domain database for analyzing the pathogenesis of OCCC to find out the differences in gene expression between OCCC and normal ovarian tissues. 

## 2. Materials and Methods

### 2.1. Microarray Datasets Gene Set Definition and Data Processing

We used the keyword of “ovarian clear cell carcinoma” and “ovary” to search for all available microarray gene expression profiles in the National Center for Biotechnology Information (NCBI) Gene Expression Omnibus (GEO) database comprehensively, and the transcriptome datasets of human OCCC and normal ovarian tissue using the Simple Omnibus Format in Text (SOFT) format were downloaded, and the detailed information of this method has been extensively reported in our previous publications [17,18,19,20,21].

The selected datasets were limited to the primary site of the ovarian tissue that had a definite diagnosis of ovarian carcinoma and normal ovarian tissue. In addition, we also excluded the gene expression profile if any missing data had been found. Based on the Human Genome Organization (HUGO) Human Genome Organization Gene Nomenclature Committee (HGNC) gene symbols approved in 2013, we performed the data analysis. After an identification of the corresponding gene symbol information in the annotation table, the microarray gene expression datasets were used. The current study included the common genes and the corresponding gene expression profiles among all datasets. If the number of the common genes became less than 8000 during intersecting with other datasets as well as the number of gene elements in the gene set was less the 3, the datasets were discarded. 

### 2.2. Detection of Differentially Expressed Genes in OCCC

To discover the DEGs (differentially expressed genes) for each of the OCCC, the aforementioned DNA microarray datasets were analyzed. We transformed and rescaled to cumulative proportion values from 0 (lowest expression) to 1 (highest expression) with an R package YuGene (version 1.1.5, downloaded from the CRAN, https://cran.r-project.org/index.html) before an integration in the gene expression levels of all samples in each dataset [22]. A linear model computed with empirical Bayes analysis by the functions “lmFit” and “eBayes” provided by the R package “limna” (version 3.26.9, downloaded from the CRAN, https://cran.r-project.org/index.html) was used to identify the DEGs. 

### 2.3. Statistical Analysis 

We performed the Mann–Whitney U test to evaluate the gene expression fold differences of the OCCC and the control groups and we corrected the results through multiple hypotheses testing using false discovery rate (Benjamini–Hochberg procedure). The significance was defined when the *p* value was < 0.01.

## 3. Results

### 3.1. Transcriptome Datasets and Gene Sets 

One hundred and eighty samples were initially collected from the GEO database, including 80 OCCC and 100 normal control samples (Figure 1). A total of 34 datasets containing five DNA microarray platforms with no missing data were integrated into the current study. Table 1 summarizes information of the samples collected. The information of the samples, including their DNA microarray platform, data set series, and accession numbers, is detailed in Appendix A.

### 3.2. Workflow

Please refer to Figure 1.

### 3.3. The Most Significant DEGs 

We identified 10,448 common genes among the five DNA microarray platforms used to detect the DEGs between the OCCC and normal ovarian control groups. Due to the large number of cases as a result of our integration of 32 datasets, 10,363 genes were found at statistical significance (adjusted *p* < 0.01). 

As listed in Table 2, the top 20 DEGs ranked by their statistical significance were RPL24, EIF3F, RPS13, EIF3L, RPS11, ITM2B, RPL27, RPL17, RPS15, RPL5, PLS3, RPS3A, RPL39, RPS27L, RPL23, RPL36AL, RPL34, ALDH9A1, RPL3, and RPL21. To protect against Akt or Myc-driven cancers in mice, partial loss of large ribosomal subunit protein 24 (RPL24) function was noted [23]. It may be partly mediated through the translational inhibition of a subset of cap-eIF4E-dependently translated mRNAs [23]. According to the function, most of these top 20 DEGs were ribosomal proteins (RPs) and eukaryotic translation initiation factors (EIFs). These two groups of genes play an important role in cancer; they are related to each other and were also largest in number in our data.

### 3.4. Summarizing the Genes Involved in OCCC Pathogenesis

The genes exhibited considerable functional commonality. Based on their statistical significance and functional commonality, we summarized the top 500 DEGs to six functional gene clusters including ribosomal proteins (RP), eukaryotic translation initiation factors (EIFs), proteasome, lactate, prostaglandin, and insulin-like growth factor (IGF) as shown in Figure 1.

The six functional gene clusters are listed in Table 3. For a more comprehensive understanding of OCCC, we reconstructed the pathogenesis network based on the six gene clusters and gene interaction database, as detailed in the Discussion section.

## 4. Discussion

In this integrated analysis, OCCC exhibited an extensive range of genetic changes and indicated its polygenic nature. These DEGs exhibited highly overlapped functionality, and they could be classified into six gene clusters. We further connected the evidence from published researches and gene interactions to reconstruct the pathogenesis network of OCCC.

### 4.1. Ribosomal Proteins Related Genes

In our data, there were 61 genes associated with ribosomal protein, which was the largest group in our classification. Most of those genes were upregulated. In addition to being involved in the processes of ribosome assembly and protein translation, ribosomal proteins were key players in regulation of apoptosis, arrest of cell cycle, cell migration and invasion, cell proliferation, and the repair of DNA damage. They also relate to tumorigenesis, immune signaling, tumor suppressors, and oncogenic signals [24].

Xu et al. discovered the tumor suppressive function of RPs, including RPS14 and RPL10; these RPs were down-regulated in our results [44]. RPS14 was reported to active TAp73, the p53 homologue, and to induce apoptosis [44]. Furthermore, RPS14 can suppress oncoproteins, such as c-Myc [24,45]. RPL10 was identified with a mutation that can function as tumor suppressor [46]. In contrast, oncogenic signals frequently promote cell growth mediated by enhancing ribosome biogenesis [47]. RPS27, RPS3A, RPL8, RPS13, RPSA, and RPL23 play different roles in relation to oncogenic signals [24,48], and the genes of these ribosomal proteins were all upregulated in our data.

The dysregulation of ribosomal proteins has already been found in some human cancers [44]. In serous epithelial ovarian cancer, low level of S4X is associated with poor prognosis [49], and L13a, L29 were noted to be upregulated in ovarian cancer [50,51]. A study indicated ribosomal protein genes play a critical role in OCCC, suggesting that these proteins could be good candidates as potential tumor markers in OCCC [52].

### 4.2. Eukaryotic Translation Initiation Factors (eIFs) Related Genes

Our data had 38 genes associated with eukaryotic translation initiation factors. All of the genes were upregulated compared with normal ovarian tissue, except for eIF3f, which exhibited down-regulation in the tissue of ovarian clear cell carcinoma.

Eukaryotic translation initiation factors were associated with the initiation phase of eukaryotic translation, cell growth, and cell cycle regulation [25]. They were reported to have an association with some diseases, such as eIF2B was noted to have involvement in neurodegenerative disease [53].

eIF4E was upregulated in breast cancer, colon cancer, head and neck cancer, non-Hodgkin’s lymphoma, and ovarian carcinoma, and was associated with the increasing grade of a disease [26,27]. The overexpressed cellular phenotype of eIF4E tends to have a more spindle-like appearance, and it can reduce cell cycle time from 20 h to 16 h. In addition, it also can induce upregulation of VEGF [25,54]. Mitogenic activity is intermediated by Ras activation [55], and c-Myc can activate eIF4E transcription [56]. Our data show agreement of the aforementioned reports, eIF4E was also upregulated in OCCC.

In melanoma and pancreatic cancer, eIF3f was down-regulated, and the overexpression of eIF3f was noted to be associated with an inhibition of proliferation and an increased apoptosis in both pancreatic cancer and melanoma cell lines [57,58]. eIF3f can also interact with heterogeneous nuclear ribonucleoprotein (hnRNP) and induce rRNA degeneration, which can inhibit protein translation [59]. Furthermore, eIF3f can suppress Akt and ERK signaling, as well as stabilize p53 [60] In our data, eIF3f was also down-regulated, as is the case in melanoma and pancreatic cancer.

### 4.3. Lactate Related Genes

Our data had four genes associated with lactate, all of which were upregulated. Some studies have demonstrated that OCCC cell lines have an increased accumulation of lactate [30,31]. Many studies have suggested that lactate has a critical role in cancer growth [28]. Furthermore, a common phenomenon in cancer cells is the Warburg effect, which is the increased uptake of glucose and accumulation of lactate [29]. Transcription factors hypoxia-inducible factor 1 (HIF-1), c-Myc oncogene, and the repression of tumor suppression factor p53 increase the expression and translocation of glucose transporters GLUT, which can enhance glucose uptake [61]. These factors increase the expression and enhance the activity of glycolytic enzymes, especially lactate dehydrogenase A (LDHA), thereby accelerating glycolysis. The downregulation of p53 also reduces mitochondrial function. Those conditions may increase lactate accumulation, and the upregulation of the monocarboxylate transporters MTC1 and MCT4 facilitate lactate exchange in candidate cancer cells. These features potentially increase the growth, proliferation, and metastasis of cancer cells [61].

Lactate can stimulate cell migration and promote metastasis. In addition, lactate can induce tumor-associated fibroblasts to secrete hyaluronan [62], a key component of the extracellular matrix (ECM), which involves the healing process [63,64,65] and also promotes metastasis of cancer cells [62]. The secretion of VEGF is induced by lactate to stand in angiogenesis [66]. There is also contribution of lactate to immune escape by interference of monocyte (inhibit the differentiation of monocytes to dendritic cells) [67,68,69], T-cell (to reduce the proliferation of T cell) [67,70], natural killercell (reduce the ability of cytolysis) [67,71], and macrophage (increase the expression of M2 macrophage) [67,72].

### 4.4. Prostaglandin Related Genes

In our data, there were six genes related to prostaglandin, most of which were upregulated. Chronic inflammation is commonly believed to contribute to the development of malignancies. It has been reported to show the strong correlation between chronic inflammation caused by infections or autoimmune diseases and cancers [73]. For example, hepatitis B virus infection may increase the risk of hepatocellular carcinoma [74,75], and high-risky types of Human papillomavirus (HPV) infection are associated with an increased risk of cervical cancer [76,77]. Patients with autoimmune diseases also have a higher incidence of malignancies, such as scleroderma and myositis [73].

Prostaglandin E2 was found to inhibit apoptosis and promote cell proliferation, migration, and angiogenesis [33]. In most human malignancies, the pathways of epidermal growth factor receptor (EGFR) [34] and COX-2 [36,37] are commonly activated. In colorectal cancer cells, it was reported that PGE2 could transactivate EGFR, which can induce cell migration and invasion by elevating PI3K-Akt signaling [78]. In addition, PGE2 can transactivate PPAR delta through PI3K-Akt pathway to inhibit apoptosis of cancer cells [79]. The expression of antiapoptotic proteins, such as Bcl-2 [80], can be induced by PGE2, and to elevate NF-kB transcriptional activity [81]. PGE2 activates Ras/Raf/MEK/ERKs pathway, which in turn increases COX-2 expression to promote cell proliferation [82]. Upregulation of IL-10 [83] and the decay accelerating factor (DAF) [84] are triggered by PGE2 to cause immunosuppression. Some studies have also demonstrated that PGE2 increases the expression of basic fibroblast growth factor (bFGF) and VEGF expression to promote angiogenesis [85].

The relation between inflammation and cancer suggests the antineoplastic activity of anti-inflammatory drugs, particularly nonsteroidal anti-inflammatory drugs (NSAIDs) [86]. As such, recent studies have demonstrated that the long-term use of NSAIDs reduces the incidence of some malignancies, including colorectal, esophageal, breast, lung, and bladder cancers [33]. Familial adenomatous polyposis (FAP), which is caused by adenomatous polyposis coli (APC) gene defects, can be treated with celecoxib (a type of COX-2 inhibitor) [87], and it can reduce the level of cancer-associated proteins, such as β-catenin, cyclin D1 [88], MMP-2, MMP-9 [89], and VEGF [90], to increase survival rate and reduce the incidence of liver metastasis.

### 4.5. Proteasome Related Genes

Proteasome is a protein complex that is responsible for the degradation of unneeded or damaged proteins. Our data had 37 genes related to proteasome, all of which were upregulated compared with normal ovarian tissue. The ubiquitin–proteasome pathway for protein degradation plays a crucial role in intracellular protein turnover. According to current studies, proteasome as a target for cancer therapy has been announced [32]. For example, bortezomib is a kind of proteasome inhibitor, which has been used to treat mantle cell lymphoma [91] and multiple myeloma [92]. The target of bortezomib is NF-κB pathway. In most cells, nuclear factor-κB (NF-kB) exists in the cytoplasm in an inactive complex bound to IkB. Cell signaling may be induced by growth factors, viruses, chemotherapy, or radiotherapy, which induce the phosphorylation and proteasomal degradation of IκB by proteasome. From such cell signaling, the transcription factor NF-κB is translocated to the nucleus to activate gene transcription and promote cell proliferation, as well as the expression of cytokines, cell adhesion molecules, and antiapoptotic proteins. Bortezomib inhibits NF-kB activity mediated through the blockage of the proteasomal degradation of inhibitor of IkB [93].

The MDM2-p53 pathway is a tumor-suppressor pathway that is often disrupted in malignancies. MDM2 is a p53-specific E3 ubiquitin ligase, which inhibits the p53 growth-suppressive function in unstressed cells. MDM2 constantly binds to p53, thereby enabling the degradation of p53 by proteasomes [94,95]. Proteasome inhibition may prevent the degradation of p53, thus activating apoptosis in cancer cells [96].

Currently, there are some molecular targets of proteasome inhibitors. In addition to NF-kB and p53, there are also the targets of p21, p27, Bax, p44/42 MAPK, tBid, and Smac/Diablo [97].

### 4.6. Insulin-Like Growth Factor Related Genes

A lot of cancers were associated with abnormal IGF signaling, such as childhood malignancies, colon cancer, melanoma, osteosarcoma, pancreatic cancer, and prostate cancer [25]. Some studies have also determined an increased IGF-1R activity to be implicated in cancer cell invasion, migration, and proliferation [42,43]. Our data had nine genes related to IGF, most of which were upregulated compared with normal ovarian tissue. Bioavailability of IGF-1 and IGF-2 is modulated by a family of IGFBPs. IGF-1R binding with either IGF-1 or IGF-2 will undergo receptor autophosphorylation and cross-linking, and then the signals activate both PI3K/Akt/mTOR and Ras/Raf/MEK/ERK pathways to enhance tumor progression [97].

In addition, it is reported that for ovarian cancer treated with either cisplatin alone or combining with paclitaxel in vitro, the development of drug resistance may be mediated through the upregulation of IGH-IGF-1R signaling pathway [98]. One study also showed the increased IGF-1R expression in ovarian cancer patients who had been treated with 3–4 cycles of the combination of platinum and paclitaxel regimen [98]. Further dissection of underlying mechanisms using a gene microarray study to evaluate 28 patients with ovarian HGSC who showed the relatively resistant to platinum chemotherapy and results had demonstrated that lots of genes involving IGF-1/PI3K/NF-kB/ERK pathway have been activated when compared to those patients who remained sensitive to the platinum-based chemotherapy [99]. In another xenograft model of ovarian HGSC, the transient knockdown of IGF-2 was noted as being able to restore paclitaxel sensitivity [100]. Ganitumab, a human monoclonal antibody to IGF-1R can enhance the therapeutic effect of platinum-based chemotherapy passing through the inhibition of the IGF-2-dependent tumor growth [101]. However, the blocking of IGF-1R cannot counteract paclitaxel resistance [76], suggesting that the development of chemo-resistance of cisplatin and paclitaxel may involve the different mechanisms.

### 4.7. Summarizing the Six Functional Gene Clusters

Figure 2 illustrates our reconstructed network of the functional gene clusters underlying the pathogenesis of OCCC.

PI3K/Akt/mTOR is often activated in malignancies, and the Ras/Raf/MEK/ERK signaling pathway is enhanced in pancreatic cancer [102], colon cancer [103], and thyroid cancer [104]. Furthermore, Ras activates PI3K/Akt/mTOR signaling [105], Akt enhances the phosphorylation of MDM2 to promote p53 degradation [106], and Akt also causes phosphorylation of IkB to enable NF-κB to translocate into the nucleus to activate genes transcription [107]. In addition, mTOR induces the release of eIF4E to begin translation initiation [108]. eIF4b also promotes c-Myc translation [109], and the release of eIF4E is elevated by c-Myc [56]. The expression of VEGF is stimulated by eIF4E [25,54], and VEGF can activate Ras to induce PI3K/Akt/mTOR pathway [110]. NF-kB is also affected by eIF1a [25]. Phosphorylation of eIF4E is promoted by ERK to activate translation initiation [25].

Upregulation of RPS3A, RPS27, and RPL8 can activate NF-kB [24]. Downregulation of RPS14 improves the expression of c-Myc [24,45] and reduces TAp73, the p53 homologue [44]. Proteasome aids in the degradation of p53 [94,95] and activation of NF-kB [93]. IGF and PGE2 can activate PI3K/Akt/mTOR, either directly or through Ras signaling [97]. PGE2 activates the Ras/Raf/MEK/ERKs pathway, which, in turn, increases COX-2 expression to promote cell proliferation. ERK increases COX-2 expression to induce the release of PGE2 [89], which stimulates VEGF expression and causes immune suppression [85]. Lactate is able to influence the NF-kB pathway, specifically in increasing VEGF [66] and inducing immune escape [79,80,81,82,83]. Hypoxia-inducible factor 1 (HIF-1), c-Myc oncogene, and p53 increase the accumulation of lactate [61].

These networks are illustrated in Figure 2, Figure 3, Figure 4 and Figure 5, where some events associated with carcinogenesis can be seen in the figures, such as angiogenesis, cell proliferation, decreased apoptosis, and immune escape. Finally, from our data, we selected some genes that are potentially related to these pathways.

## 5. Conclusions

OCCC is a chemoresistant subtype that is associated with poor prognosis. Therefore we tried to determine the associated mechanism in the hope that we can find a better treatments. We used integrated analysis, and based on the commonality and current studies, we selected six functional gene clusters that may be involved in the pathogenesis of OCCC, including ribosomal protein, eukaryotic translation initiation factors, lactate, prostaglandin, proteasome, and insulin-like growth factor. Based on the close interaction among these clusters, and in conjunction with findings in the literature, we further reconstructed the network of OCCC pathogenesis. According to the results, we may provide insight or targets for therapeutic strategies in the future.

## Figures and Tables

**Figure 1 ijerph-17-03951-f001:**
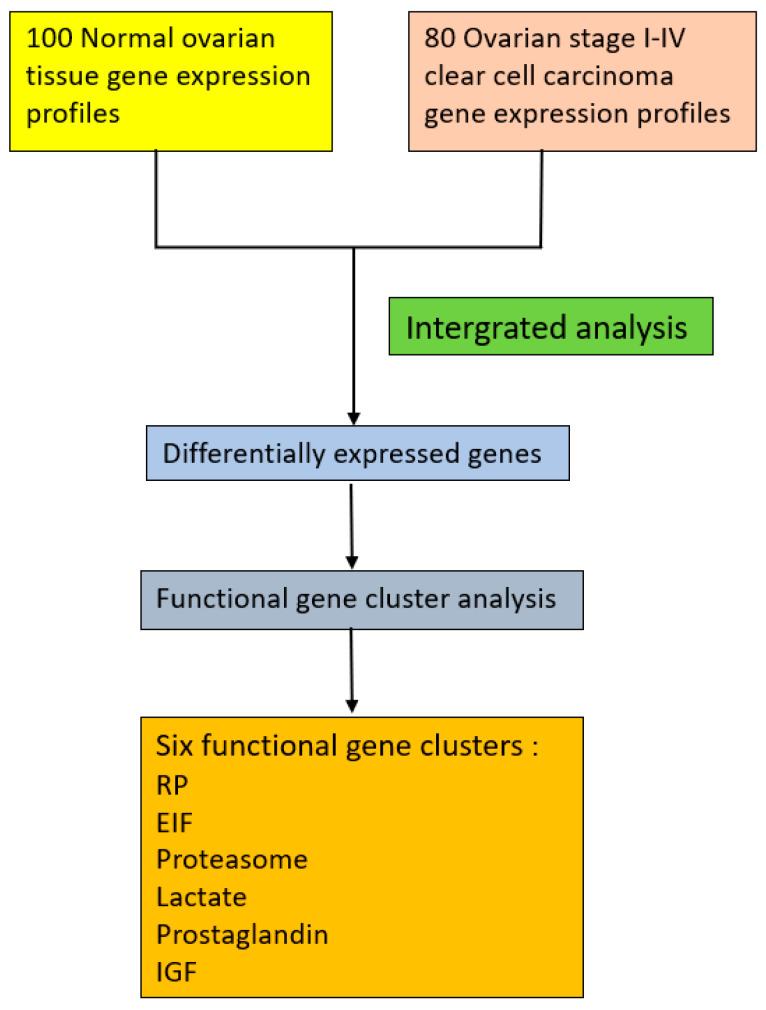
Workflow for selecting the six functional gene clusters. RP: ribosomal protein; EIF: eukaryotic translation initiation factor; IGF: Insulin-like growth factor.

**Figure 2 ijerph-17-03951-f002:**
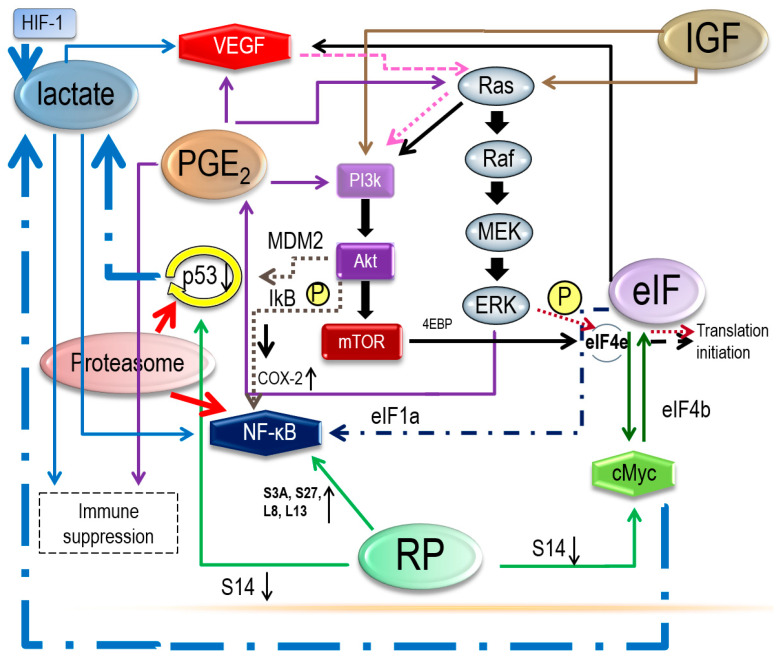
Associations between ribosomal proteins, eukaryotic translation initiation factors, lactate, prostaglandin, proteasome, and insulin-like growth factors, as detailed in the Discussion section.

**Figure 3 ijerph-17-03951-f003:**
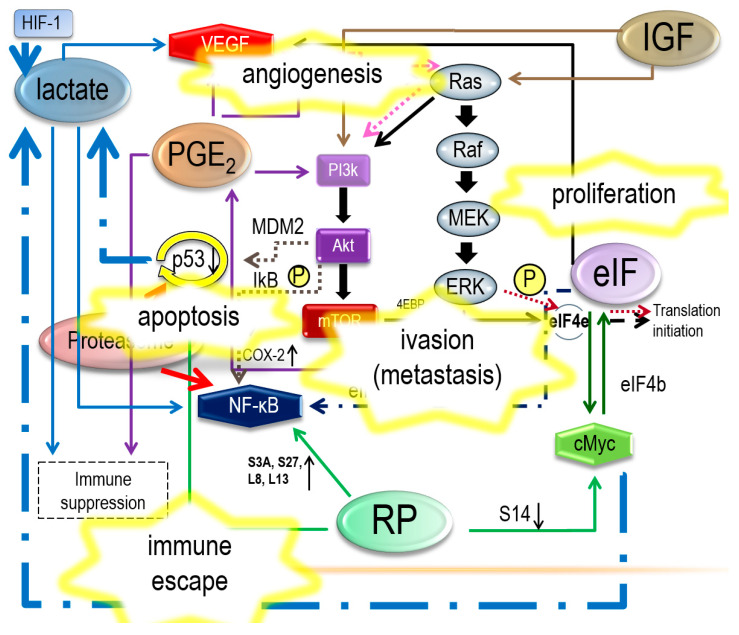
Some events associated with carcinogenesis: angiogenesis, cell proliferation, decrease of apoptosis, and immune escape; these appear to match some of the conditions necessary for carcinogenesis.

**Figure 4 ijerph-17-03951-f004:**
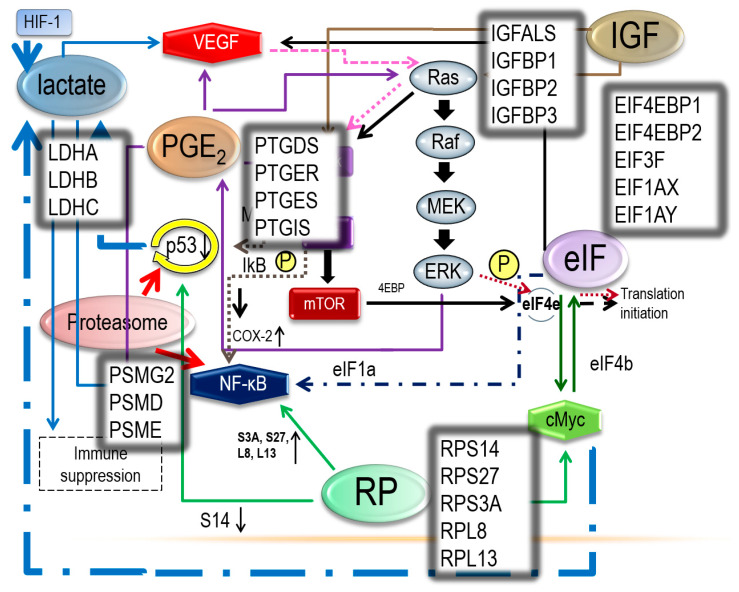
We identified the commonality (Figure 2) based on the six functional gene clusters. Using findings from the literature, we selected several genes in each functional gene cluster that are more likely to be related to the carcinogenesis of OCCC.

**Figure 5 ijerph-17-03951-f005:**
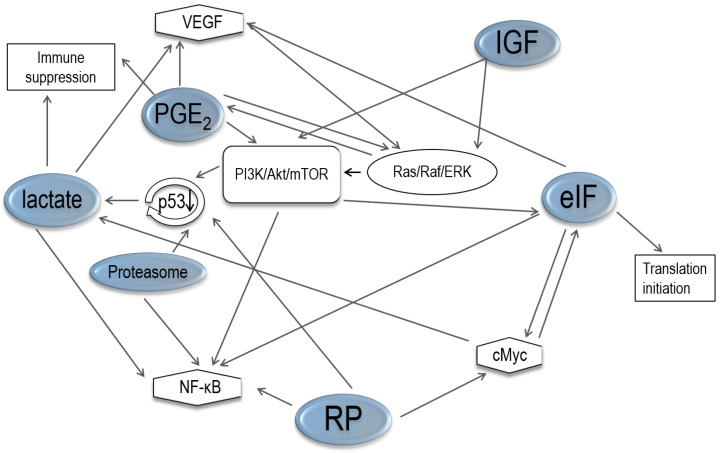
Relationships between ribosomal proteins, eukaryotic translation initiation factors, lactate, prostaglandin, proteasome, and insulin-like growth factors.

**Table 1 ijerph-17-03951-t001:** Microarray platforms in our samples.

Microarry. Plateform	Case Numbers of OCCC (Homo Sapiens)	Case Numbers of Normal Ovarian Cell (Homo Sapiens)
GPL 570	49	95
GPL 6244	12	0
GPL 96	11	0
GPL 6947	5	5
GPL 7264	3	0
Total numbers	80	100

**Table 2 ijerph-17-03951-t002:** Top 20 DEGs in our data.

Top 20 DEGs	*p*-Value
1. RPL24	1.23 ×10^−109^
2. EIF3F	7.35 × 10^−109^
3. RPS13	9.3 × 10^−102^
4. EIF3L	1.52 × 10^−101^
5. RPS11	1.71 × 10^−98^
6. ITM2B	6.57 × 10^−98^
7. RPL27	7.35 × 10^−98^
8. RPL17	5.33 × 10^−97^
9. RPS15	2.73 × 10^−94^
10. RPL5	2.97 × 10^−92^
11. PLS3	1.17 × 10^−91^
12. RPS3A	1.42 × 10^−91^
13. RPL39	7.65 × 10^−91^
14. RPS27L	7.68 × 10^−90^
15. RPL23	9.84 × 10^−90^
16. RPL36AL	1.60 × 10^−89^
17. RPL34	1.98×10^−89^
18. ALDH9A1	2.31 × 10^−89^
19. RPL3	6.82 × 10^−89^
20. RPL21	9.70 × 10^−89^

**Table 3 ijerph-17-03951-t003:** Six functional gene clusters identified in our data: ribosomal protein (RP), eukaryotic translation initiation factors (EIFs), lactate, proteasome, prostaglandin, and insulin-like growth factor (IGF).

Gene Group	Genes	*p*-Value	Function of the Genes
Ribosomal protein (RP)	RPL24, RPS13, RPS11, RPL27, RPL17, RPS15, RPL5, RPS3A, RPL39, RPS27L, RPL23, RPL36AL, RPL34,RPL3, RPL21, RPL36, RPL30, RPL6, RPL32, RPL31, RPS16, RPL13A, RPS20, RPS12, RPL41, RPS4X, RPSA, RPL18, RPS24, RPL35A, RPS17, RPL13, RPL10A, RPS27, RPS6, RPS18, RPL27A, RPS27A, RPL23A, RPS28, RPS26, RPS2, RPLP2, RPL35, RPS21, RPL8, RPS5, RPL11, RPL10, RPL7, RPS14, RPL7A, RPL14, RPL37, RPL26, RPN2, RPL26L1, RPS6KA2, RPRD2, RPS6KC1, RPRM, RPL10L, RPP40, RPIA, RPP14, RPS6KA3, RPS6KB1, RPP38, RPP25, RPS6KA5, RPP30, RPS6KA6, RPS6KB2, RPL39L, RPS4Y1, RPL3L, RPS6KA1, RPS6KA4	1.23 × 10^−109^ ~ 2.85 × 10^−16^	Protein translation. Regulation of apoptosis, cell cycle arrest, cell proliferation, cell migration and invasion, DNA damage repair. Tumorigenesis, immune signaling, tumor suppressors, and oncogenic signals [24].
2. eukaryotic translation initiation factors (eIF)	EIF3F, EIF3L, EIF3E, EIF1, EIF4B, EIF4A2, EIF2S3, EIF4H, EID1, EIF3G, EIF3M, EIF3K, EIF4A3, EIF5, EIF3A, EIF5A, EIF2AK1, EIF1AX, EIF3C, EIF2S2, EIF24, EIF3B, EIF2B1, EIF2B2, EIF6, EIF4EBP2, EIF2B4, EIF5B, EIF2S1, EIF2AK2, EIF4ENIF1, EIF3J, EIF2B5, EIF4G3, EIF4EBP1, EIF5A2, EIF2AK3, EIF1AY	1.52 × 10^−101^ ~ 0.00412202	Initiation phase of eukaryotic translation, cell growth and cell cycle regulation [25]. In some malignancies: up-regulation of eIF4e. Associate with increasing grade of disease [26,27].
3. Lactate	LDHA, LDHB, LDOC1, LDLRAP1, LDLR, LDHC, LDHAL6B	1.38 × 10^−72^ ~ 9.07 × 10^−06^	Cancerous growth [28] Warburg effect [29] Increases accumulation of lactate in OCCC cell line [30,31].
4. Proteasome	PSMB4, PSMG2, PSMC1, PSMB6, PSMA4, PSMB5, PSME1, PSMD4, PSMC2, PSMD14, PSMB3, PSMB7, PSMA5, PSMC5, PSMC6, PSMD1, PSMD10, PSMB2, PSMA2, PSMD5, PSMD9, PSMD8, PSMD12, PSME4, PSMA3, PSMD6, PSMD7, PSME3, PSMB8, PSMB9, PSMC4, PSMD11, PSMB10, PSMF1, PSMD13, PSMC3IP, PSMD3	3.30 × 10^−70^ ~ 1.59 × 10^−18^	Degradation of unneeded or damaged proteins. Ubiquitin-proteasome pathway: intracellular protein turnover. As a target for cancer therapy [32].
5. Prostaglandin	PTGIS, PTGES3, PTGS1, PTGER4, PTGDS, PTGES, PTGER2, PTGES2, PTGIR, PTGDR, PTGS2, PTGFR, PTGER3, PTGER1	1.60 × 10^−66^ ~ 1.80 × 10^−13^	Prostaglandin E2 promotes cell proliferation, migration, and angiogenesis, and inhibition of apoptosis [33]. In most human malignancies: epidermal growth factor receptor (EGFR) [34,35,36] and COX-2 [37,38,39,40] are activated commonly.
6. Insulin-like growth factor (IGF)	IGFBP3, IGFBP6, IGFBP5, IGFBP2, IGFALS, IGFBP1	2.72 × 10^−57^ ~ 5.00 × 10^−08^	Numerous human cancers abnormal IGF signaling [38]. Increased IGF-1R activity: cancer cell proliferation, migration, and invasion [41,42,43].

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
