# Peer review of "Functional Gene Clusters in Global Pathogenesis of Clear Cell Carcinoma of the Ovary Discovered by Integrated Analysis of Transcriptomes"

_ijerph, 2020, doi:10.3390/ijerph17113951_

Round 1

Reviewer 1 Report

Authors proposed functionnal network for OCCC based on retrospectival intagrated transcriptomic analysis.

Concerning Figures2, 3, 4, 5: authors should be more precise and explain nature of links and arrows (gene physical interaction, protein interaction, gene expression link?). If not, authors should explain why they use different colors (each color of links correspond to the gene involved).

Maybe in conclusion, authors should include links between the functionnal network they show and therapeutics for ovarian cancers and/or proposed new therapeutic targets (protein? signaling pathway?)

Author Response

Thanks for your suggestion and our replies is in the attachment.

Reviewer 2 Report

Review for Manuscript ijerph-800682-peer-review-v1

General Comments: This study interrogated gene expression datasets that are available to the public to identify gene/gene families implicated in OCCC pathogenesis. A few general comments:

  • The manuscript needs English editing.
  • Line 19 – “order to find the pathogenesis involved in OCCC carcinogenesis”. From these types of studies, you cannot say causation of pathogenesis. This is only possible with gain or loss of function studies looking at one or multiple genes potentially involved using cell lines and animal models. It should be betters stated that this study determined genes that may be implicated in OCCC pathogenesis.

 More Specific Comments:

Title – None

Abstract

  • See comment #2 above

Introduction

  • Line 47 – “and so on” should not be used. If a pathway is not important, it should not be listed. If it is important, it should be listed.
  • Line 52-53 – The BET inhibitor sentence should be removed, since this is not a treatment study.

Materials and Methods

  • Line 90 – Insert “testing” after “hypotheses”

Results – None

Discussion

  • Line 149 – Change “nature of a polygenic disorder” to “polygenic nature”
  • Line 193 – Change “about lactate” to “associated with lactate”
  • Line 239-240 – The interest in COX-2 inhibitors and colon cancer sentence should be removed, since it is not relevant to this study.

Figures, Tables, and Legends – Very good

Author Response

(The authors gave the same response as above.)

Round 2

Reviewer 2 Report

Review for Manuscript ijerph-800682-peer-review-v1

General Comments: The authors have nicely and fully addressed all of my comments. No further questions or comments.